# An Improved Method for Quantification of Viable *Fusarium* Cells in Infected Soil Products by Propidium Monoazide Coupled with Real-Time PCR

**DOI:** 10.3390/microorganisms10051037

**Published:** 2022-05-17

**Authors:** Lida Chen, Lei Li, Xuewen Xie, Ali Chai, Yanxia Shi, Tengfei Fan, Jianming Xie, Baoju Li

**Affiliations:** 1College of Horticulture, Gansu Agricultural University, Lanzhou 730070, China; 15369301173@163.com; 2Institute of Vegetables and Flowers, Chinese Academy of Agricultural Sciences, Beijing 100081, China; caulilei@163.com (L.L.); xiexuewen@caas.cn (X.X.); chaiali@163.com (A.C.); shiyanxia@caas.cn (Y.S.); fantengfei@caas.cn (T.F.); 3Shouguang R&D Center of Vegetables, Chinese Academy of Agricultural Sciences, Weifang 262700, China

**Keywords:** *Fusarium*, cucumber, propidium monoazide (PMA), real-time PCR, viable cells

## Abstract

*Fusarium* is a soil-borne pathogen that causes root rot disease in cucumber. To date, quantitative real-time PCR (qPCR) is a common tool to detect the content of *Fusarium* in soil. However, qPCR cannot distinguish between viable and nonviable cells. The aim of this study was to develop a detection technique to pretreat tissue fluid with propidium monoazide (PMA) followed by extract DNA, and then to quantify viable *Fusarium* cells in contaminated soil. In this work, the specific primer pair F8-1/F8-2 was designed based on the translation elongation factor (EF) gene and a PMA-qPCR assay was established to amplify and quantify soils of viable *Fusarium* cells. The PMA pretreatment test was optimized, which indicated that the optimal PMA concentration and light exposure time were 50 mmol L^−1^ and 15 min, respectively. The lowest limit of viable cells in suspension detected and soil by PMA-qPCR were 82 spore mL^−1^ and 91.24 spore g^−1^, respectively. For naturally contaminated soil, viable *Fusarium* cells were detected in eight of the 18 samples, and the *Fusarium* amount ranged from 10^4^ to 10^6^ spore g^−1^. In conclusion, the PMA-qPCR method has the characteristics of high sensitivity, efficiency, and time saving, which could support nursery plants to avoid *Fusarium* infection and agro-industry losses.

## 1. Introduction

*F**usarium* sp. are fungi that are the causal agents of vegetable root rot, which is an economically important soil-borne disease of vegetable species and is responsible for substantial crop losses [1]. *F**usarium* sp. include *F**usarium solani*, *F**usarium*
*oxysporum*, *Fusarium equiseti,* and *Fusarium moniliforme*, which can infect a variety of crops and vegetables [2].

Cucumber (*Cucumis sativus* L.) is an important global vegetable crop whose production can be severely hindered by infection with *F**. solani*. This pathogen causes root rot disease with symptoms that include necrotic lesions and vascular and root rot, threatening cucumber production around the world [3]. *F. solani* was first discovered in South Africa in 1930 and subsequently reported in different areas, such as Africa and China [4]. *F. solani* often survives in soil or diseased bodies for a long time, as it can form resistant structures called chlamydospores [5]. These characteristics make eradication quite difficult because chlamydospores, conidia, and mycelia may be sources of infection [6]. It is obviously necessary to perform the rapid and accurate detection of *F. solani* to ensure the healthy growth of cucumbers and reduce the incidence of disease.

So far, methods to detect *F. solani* are based on pathogen isolation, microscope observation, polymerase chain reaction (PCR), real-time qPCR (qPCR), TaqMan real-time PCR, amplified fragment length polymorphisms (AFLPs), and others [7,8,9,10]. However, the limitation of nucleic acid-based detection techniques is that they cannot distinguish between viable and nonviable cells, since the DNA of dead cells can exist in air or soil for a long time [11]. Hence, with the DNA of dead cells, PCR amplification overestimates the number of viable *Fusarium* sp. cells. In addition, they are potential mycotoxin producers. Therefore, these detection methods cannot be used to assess the potential risk of fungal infection in plant products or soil.

To overcome this problem, PMA coupled with qPCR has been proven to be applicable to the detection of dead and viable cells. PMA is one of the most frequently applied non-membrane-permeating dyes, since PMA dye penetrates only cells with damaged membranes and not cells with intact membranes [11]. Therefore, PMA can inhibit the PCR amplification of dead cell DNA and reduce the overestimation of cell count caused by dead cell DNA in qPCR detection [12]. Previous studies have shown that PMA-qPCR technology can be used for detecting and quantifying viable bacterial cells in different vegetable industry fields [13,14,15,16,17,18,19,20] and specific microorganisms in some food industries [21,22]. However, few reports have focused on fungi [23,24], and no studies have involved *Fusarium*. Here, it is necessary to develop an effective PMA-qPCR method to detect the viable cells of *Fusarium* in soil samples.

Therefore, the study aim was to (i) design specific primers for *Fusarium* based on the translation elongation factor (EF) gene sequences; (ii) optimize the treatment concentration and light exposure time of PMA; (iii) detect the sensitivity of *F. solani* in the soil; and (iv) estimate the sensitivity of the PMA-qPCR on infected soil in Shandong, Hebei, and Beijing provinces.

## 2. Materials and Methods

### 2.1. Fungal Isolates, Cultivation, and DNA Extraction

In total, 19 strains, including 9 *F**usarium* sp. strains and 10 non-target strains, were used for primer development and species detection (Table 1). Cultures were grown on Potato Dextrose Agar (PDA) medium at 26 °C for 5 days in the dark. Genomic DNA was extracted from fungal mycelia grown on PDA plates using the CTAB method according to the TIANamp Soil DNA Kit instructions [25]. Conidia were collected with a sterile solution of Tween 80 (0.005% *v*/*v*) and filtered through Miracloth (Calbiochem, CA, USA). The DNA concentration was determined by a Thermo Scientific Nanodrop 2000 spectrophotometer (Wilmington, DE, USA), and the samples were stored at −40 °C for future analysis.

### 2.2. Primer Design and qPCR Conditions

In this study, by downloading the partial sequence of the *Fusarium* EF gene (MK077042) as available in the NCBI database, and using MEGA 7.0 software to compare the sequences, a total of 15 sequences (Table 2) were designed.

The qPCR detection procedure was consistent with the method described by Chai et al. (2020) [26]. Briefly, the 20 μL reaction volume containing 10 μL of SuperReal PreMix Plus, 0.2 μL of F8-1/F8-2 primer (10 μM), 1 μL of template DNA (210 ng) and 0.4 μL of 50 × ROX Reference Dye. Amplification was performed by using the ABI 7500 Real-Time PCR System (Applied Biosystems, Carlsbad, CA, USA). Fluorescence was detected after each cycle. Cycle threshold (Ct) values were calculated automatically by ABI 7500 software (Applied Biosystems, Carlsbad, CA, USA).

### 2.3. Establishment of Standard Curve

The dilution of *F. solani* with sterile water in a 10-fold dilution series from 8.2 × 10^7^ to 82 spore mL^−1^ and subsequent DNA extraction and qPCR were performed as described above. The concentration of each sample was repeated for 3 times to determine the standard deviation (SD) of Ct value. The coefficient of variation (CV) was calculated following a method similar to Chai et al. (2020) [26]. It is noteworthy that the limitation of quantification was specified as the lowest concentration at which replicates showed a CV ≤ 35% at the calculated concentration [27].

### 2.4. PMA Treatment

PMA treatment was optimized by using *Fusarium* suspension with different concentrations of viable cells. Tubes containing 1 mL of 10^7^ spore mL^−1^ suspensions were heat inactivated at 100℃ for 10 min to prevent cell survival [28]. Then heat inactivation suspension was cooled to room temperature, and 100 µL was extracted and placed on 27 °C PDA for 96 h to confirm that the cells were dead.

Before the test, 20% dimethyl sulfoxide (DMSO) was used to dissolve the PMA solution to the concentration of 20 mmol L^−1^ and it was stored at −20 °C (dark environment). Under the condition of weak light, the dissolved PMA was quickly added to the sample tubes containing 200 μL of viable/dead cells (derived from 10^7^ spore mL^−1^ suspensions) to make the final concentrations of 0, 20, 30, 40, 50, 60, and 70 μmol L^−1^. After incubating in the dark for 30 min, a 50 W LED light source was used to illuminate sample tubes at a distance of 15 cm for 15 min according to aprevious method [23]. A 200 μL aliquot of the suspension was boiled for 10 min at 100 °C, and after centrifugation (12,000× *g*, 1 min). The DNA was extracted using a TIANamp Fungi DNA Kit (Tiangen Biotech Co., Ltd. Beijing, China), and in final 1 μL DNA was used directly in qPCR. In order to determine the optimal light exposure time to inhibit *F. solani* viable cell suspensions, 10 mmol L^−1^ PMA solution was added to the sample tube of viable or dead cell suspensions in different proportions. The light exposure time was set as 0, 5, 10, 15, 20, and 25 min. All samples were centrifuged at 10,000× *g* for 1 min, and DNA was extracted and amplified by qPCR as described above after PMA treatment. The PMA-qPCR detection of each sample involved the preparation of 3 replicates to ensure the repeatability and reliability of the test.

### 2.5. Detection of Defined Ratio of Mixture Suspension

In order to clarify the applicability of the PMA-qPCR method for distinguishing viable and dead cells of *F. solani*, different proportions of mixture suspensions (0.1%, 1%, 25%, 50%, 75% and 100% viable spore) with a concentration of 1 × 10^7^ spore mL^−1^, were treated with optimized PMA concentration. Different proportions of mixed suspension (200 μL each) were tested by three methods. (i) mixed suspension diluted to 10^2^ spore mL^−1^ with sterile water, before the extraction of 100 μL for PDA medium to count fungal colonies (spore)l (ii) DNA from the mixed suspension was directly extracted and amplified by qPCR; and (iii) DNA was extracted from the mixed suspension after PMA pretreatment and amplified by qPCR. Each treated sample was repeated in 3 replicates.

### 2.6. Detection of F. solani from Artificially Contaminated Soil

The soil was sterilized at 121 °C for 20 min. Then, suspensions of spore of *F. solani* were treated in distilled water containing Tween 80 (0.005% *v*/*v*). The soil (200 g) was infected with 200 mL of spore suspensions (10^7^, 10^6^, 10^5^, 10^4^, 10^3^, 10^2^, and 10 spore g soil) and homogenized for 60 s. Another 200 g sterilized soil not inoculated with *F. solani* and infected soil with 10^7^ dead spores was used as the negative control. To analyse the infection of soil *F. solani* by three methods: (i) 9 mL sterile water and 1 g treated soil were mixed, and 100 μL extracted for PDA medium to count fungal colonies (spore); (ii) pellets were re-suspended in 2 mL of phosphate buffer saline (PBS), before DNA was extracted after treatment with optimized PMA; and (iii) for bioassays, the soil (200 g) containing *F. solani* was placed in sterilized plastic flowerpots with a diameter of 10 cm. The 50 healthy cucumber seeds were randomly selected and sown in flowerpots for growth in greenhouse. The greenhouse ambient temperature of 18~20 °C (night) and 26~30 °C (day) and 80~100% relative humidity (RH) to checked for the presence or absence of cucumber fungal pathogens. Symptoms of cucumber root rot were checked after 7 days. Disease incidence (DI) was calculated as the percentage of plants with symptoms caused by *F. solani* in each treatment. Only disease incidence was recorded because commercially diseased plants at any level of severity were discarded at the nurseries.

### 2.7. Detection of F. solani from Naturally Infested Soil

In 2019–2020, 18 soil samples (about 1 kg) were collected in *F. solani*-infested fields of Shandong, Hebei, and Beijing provinces. The soil was sifted to remove excess plant residues. 200 g of soil sample and 200 mL of sterile distilled water were mixed in equal proportions, then incubated in a shaking table for 10 min (25 °C in the dark). The treatment of soil suspension shall refer to the description of seed suspension by Chai et al. (2020) [26]. Each sample was verified by bioassays and PMA-qPCR.

### 2.8. Data Analysis

The differences of various detection techniques were compared using the Analysis of Variance (ANOVA, IBM SPSS 20.0). The significance value was determined at the *p* < 0.05 level and at the 95% confidence level by the Tukey test. Ct values and infection rates were expressed as the mean ± SD.

## 3. Results

### 3.1. Preliminary Screening of the Primers

The instructions of the soil DNA extraction kit, in which only the F8-1/F8-2 primer pair amplifies a single fragment of 189 bp, were used for PCR. The other primers did not amplify a good single band in the corresponding region (Figure 1). Therefore, in the primer screening test, only primers F8-1/F8-2 were used as targets for the next test. A primer pair, F8-1 (5′-GCTTCTCCCGAGTCCCA-3′) and F8-2 (5′-GCTCAGCGGCTTCCTAT-3′), was designed for the specific amplification of Fusarium DNA based on the sequence alignment of EF gene sequences from *Fusarium* strains reported in a previous study and reference strains in the NCBI database (Figure 2).

### 3.2. Specificity of the Primers

A PCR assay confirmed the positive samples and a predictable fragment (189 bp) was observed on agarose gel. Sequence comparison showed that the amplified sequence was 100% similar to the *F. solani* sequence from the GenBank database. In this sense, the primer can amplify the correct product and has specificity for *Fusarium*. The primers were specific to *Fusarium*, of which nine species of *Fusarium* were amplified, and the other 10 fungal strains were not amplified by qPCR (Figure 3A). In addition, we found that the optimal annealing temperature of primers for qPCR detection of *Fusarium* was 60 ℃. The Ct values of all *F. solani* strains were amplified between 18.37~26.52, which were valid data (Table 1). The melting curve of *Fusarium* strains had a single peak with a Tm = 87, while other fungal strains had no peak (Figure 3B). In this sense, those samples with Ct > 35 and a double peak on the melting curves were considered invalid for the quantification of *F**usarium*.

### 3.3. Sensitivity and Standard Curve of qPCR

Using the *F. solani* suspension with an initial concentration of 8.2 × 10^7^ spore mL^−1^, seven concentrations were diluted according to a 10 fold gradient and a quantitative standard curve was constructed. The concentration of *F. solani* spore suspensions has a good linear relationship with the corresponding Ct values (y = −3.1388x + 40.166). The correlation coefficient (*R*^2^) was 0.9985 (Figure 4A). The dilution of pure *F. solani* spore suspensions at a concentration of 82 spore mL^−1^ produced replicates with a Ct of 34.53. A lower concentration of *F. solani* spore suspension (8.2 spore mL^−1^) produced replicates with a Ct > 35. Therefore, the sensitivity limit of qPCR for the detection of *F. solani* suspension was 82 spore mL^−1^ (Figure 4B).

### 3.4. PMA Concentration Optimization

Viable and dead cells of *F. solani* were pretreated in different concentrations of PMA (0, 20, 30, 40, 50, 60, and 70 μmol L^−1^). It is noteworthy that the Ct values of heat inactivated *F. solani* spores were slightly higher than viable cells. For dead *F. solani* cell suspensions, the Ct values increased significantly with the increase of PMA concentration from 0 to 50 mmol L^−1^ (*p* < 0.05) (Figure 5A). When the PMA concentration increased from 50 to 70 mmol L^−1^, there was no significant difference in Ct value. Therefore, 50 mmol L^−1^ was considered the minimum optimal PMA concentration. For viable cell suspensions, there was no significant difference in Ct value with the increase of PMA concentration from 0 to 70 mmol L^−1^ (*p* < 0.05). To sum up, PMA concentration had no effect on DNA amplification of viable cells.

The trend of light time screening and concentration treatment was almost similar. The light time presented no significant difference in the Ct values of viable *F. solani* cells (*p* < 0.05) (Figure 5B). The Ct values showed a significant upward trend with PMA exposure time from 0 to 15 min for dead cell suspensions (*p* < 0.05). When the light exposure time was extended from 15 to 25 min, the Ct value did not change significantly (Figure 5B). Thus, 15 min will be the appropriate exposure time to completely inhibit the DNA amplification of dead cells.

### 3.5. Validation of PMA-qPCR

The PMA-qPCR assay, qPCR assay, and plate-counting method were used to detect the mixed suspension in different proportions. There was no significant change in different concentrations of cells by qPCR. In contrast, the PMA-qPCR and colony counting method showed significant differences in the detection of different proportions of *F. solani* suspension. When the ratio of viable cells decreased from 100% to 0.1%, the Ct values significantly increased from 15.45 ± 0.45 to 29.09 ± 0.18, and the corresponding viable *F. solani* cell counts decreased significantly from 7.87 ± 0.14 to 3.53 ± 0.06 log spore mL^−1^ (*p* < 0.05). Colony counting analysis showed that with the decrease in the ratios of viable spores, the spore count gradually decreased from 7.03 ± 6.98 to 4.82 ± 0.47 log spore mL^−1^ (Table 3).

### 3.6. Detection of Soil Inoculated with F. solani by Different Methods

Different concentrations of viable *F. solani* spore (10~10^7^ spore mL^−1^) and dead spores (10^7^ spore mL^−1^) were used for determination using PMA-qPCR, plate-counting assay, and bioassays, respectively.

The highest number of viable *F. solani* spores was detected in soil samples contaminated with a fungal suspension of 10^7^ spore mL^−1^. The content of *F.*
*solani* in soil was detected by the PMA-qPCR method and plate-counting method, which resulted in 8.77 × 10^6^ spore g^−1^ and 9.13 × 10^6^ spore g^−1^, respectively. With the decrease of the number of viable cells in soil infected with *F. solani*, the detection values of PMA-qPCR and plate-counting decreased gradually. The soil was fairly lowly infected with a *F.*
*solani* suspension of 10^2^ spore mL^−1^, with about 91.24 spore g^−1^ of soil using PMA-qPCR, while no viable *F. solani* cells were detected in soil samples using the plate-counting assay. Therefore, the PMA-qPCR assay was more sensitive in the detection of soil viable *F. solani* (Table 4).

In the bioassays, soil inoculation of *F. solani* was positively correlated with cucumber incidence rate. In the treatment of inoculating 10^7^ viable spores of *F. solani* in soil, the incidence of cucumber root rot was the most serious, and its DI was 92.67 ± 2.62. We found that the infestation levels of 10^5^~10^6^ viable spores of soil gave DI values of 47.33 ± 1.25 and 76.00 ± 2.45 (*p* < 0.05), respectively. In contrast, the lowest infection level of symptoms of cucumber root rot was 10^4^ viable spores, and its DI was 10.67 ± 1.25. However, no symptoms of cucumber root rot were observed in soil samples with infection levels less than or equal to 10^3^ viable spores (Table 4).

### 3.7. Detection of F. solani in Naturally Infested Soil

The PMA-qPCR assay demonstrated that the viable *F. solani* spores could be reliably quantified in eight naturally infested soil samples, with the soil *F. solani* content ranging from 10^4^ to 10^6^ viable spores. The highest levels of viable *F. solani* spores were detected in soil samples originating from an *F. solani*-infested field in Shandong Province (samples 3, 4 and 5), with Ct values ranging from 18.25 ± 0.30 to 21.38 ± 0.27, which is equivalent to 9.60 × 10^6^ to 9.66 × 10^5^ spore g^−1^ by PMA-qPCR. Viable *F. solani* spores were also found in soil samples harvested from Beijing city (samples 1 and 2) and Hebei Province (samples 6–8), which showed Ct values of 23.87 ± 0.27 to 26.75 ± 0.22, corresponding to approximately 1.56 × 10^5^ to 1.88 × 10^4^ spore g^−1^ (Table 5).

In the bioassays, cucumber plants in eight of 18 soil samples showed symptoms of root rot, and no symptoms were observed in other soil samples (Table 5). In the soil treatment containing 10^5^~10^6^ spore g^−1^ in Shandong Province, it was found that the incidence of cucumber root rot was more serious, and its DI > 55 (samples 3~5). The incidence of cucumber root rot was 32.67~35.33 in Beijing (samples 1~2). In addition, the incidence of cucumber root rot was light, and its DI was 14.67~16.00 in Hebei (samples 6~8). The other 10 soil samples showed no typical symptoms of cucumber root rot, and the Ct value amplified by PMA-qPCR was >35. Therefore, the PMA-qPCR method could be applied to detect the content of viable cells infected with *F. solani* in natural soil in different areas.

## 4. Discussion

The PMA-qPCR technology has been applied to the detection of plant pathogen cells [26,29,30]. Despite the fact that *F*. *solani* can be identified by qPCR, it has the disadvantage of being unable to distinguish between viable cells and dead cells in plant or soil. In this research, the quantitative detection method of PMA combined with qPCR is established for the first time, which can be used for the monitoring of naturally infected soil diseases and is of great significance for the early warning of crop soil-borne diseases.

In this research, the optimal PMA concentration and light exposure time are 50 mmol L^−1^ and 15 min, respectively, to distinguish between dead and viable *F. solani* spore. When the PMA concentration is 10 μg mL^−1^, exposed for 10 min, the amplification of *Pseudomonas syringae* pv. *maculicola* DNA can be inhibited [31]. Previous studies reported that PMA treatment is 3 μg mL^−1^ to inhibit the PCR amplification of DNA from *Bacillus cereus* dead cells [32]. It is reported that the preparation of 1 mg mL^−1^ PMA with an optimized volume of 10 µL and illumination for 5 min can effectively detect the content of viable cells of *Escherichia coli* [33]. In addition, there are many reports about optimizing the PMA system to detect soil pathogens [34,35]. Among the studies of soil-borne pathogens, there is inconsistency regarding the optimal PMA concentrations or light exposure times, which may be the target fungi.

The EF gene is a conserved gene of the *Fusarium* sequence, and the specific DNA sequence of *Fusarium* can be detected. The assay detected only DNA sequences specific to *Fusarium* sp. No signal is obtained for ten other fungal species (Figure 3A). However, it is not possible to distinguish between *F*. *solani* and *Fusarium* sp. using F8-1/F8-2 primers, but the ability to detect more than two *Fusarium* sp. may be an advantage in the diagnosis of soil borne diseases in the vegetable industry, because of their related mycotoxigenic profile. Previously, the primer sequences used to detect *F. solani* were derived from nuclear ribosomal RNA genes [36], but the EF marker has been demonstrated to be more specific for species discrimination [37].

In the test, using the established PMA-qPCR method, the limit of viable cells in artificially infected soil *F. solani* suspension is 91.24 spore. The PMA-qPCR assay demonstrated that the limit of detection (LOD) of the assay reached 10^2^ CFU mL^−1^ of *Staphylococcus aureus* in spiked milk [38]. For the activity detection of *Bacillus cereus* in the food field, the detection limit of the PMA-qPCR method is 7.5 × 10^2^ CFU mL^−1^ [39]. For *Alternaria* spp. in the vegetable industry, the LOD of detected by PMA-qPCR method is 10^2^ spore g^−1^ of tomato [24]. In the bioassay, the infection level of soil with low incidence of cucumber root rot is 4.10 × 10^3^ spore g^−1^, with a DI of 10.67. Typical symptoms of soil-transmitted cucumber root rot disease are found in soil samples at a degree of infection greater than 10^3^ spore g^−1^. When the content of *F. solani* infection in soil is less than or equal to 10^3^ spore g^−1^, cucumber plants grow healthily and do not suffer from root rot. In addition, the detection results of PMA-qPCR are agreement with those of traditional separation methods. Among the discrepancies, PMA-qPCR can detect 100 spores in the soil tested, but not plant-counting assays, which proves that the PMA-qPCR detection method demonstrates higher sensitivity and is better than the plant-counting method. The PMA-qPCR assay shows that eight of the 18 natural soil samples carried *F. solani* viable cells, indicating that it may lead to the occurrence of cucumber plant root rot in naturally infected soil.

In conclusion, this study presents a rapid, sensitive, and reliable tool for the quantitative detection of soil-borne viable *Fusarium* fungi, that can be applied to produce accurate and reliable data for risk assessments in vegetable health. The use of this assay will allow to carry out the preventive detection of soil before plant transplantation to field production. So far as we know, the application of the PMA-qPCR assay for the quantification of *F.*
*solani* viable spores in naturally infected soil has not been described to date.

## Figures and Tables

**Figure 1 microorganisms-10-01037-f001:**
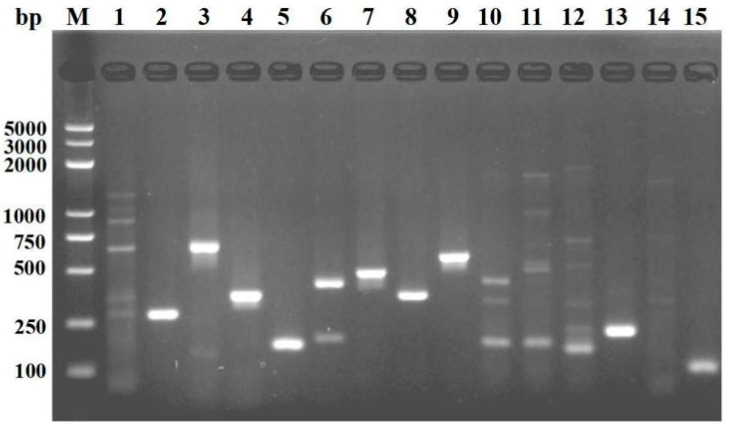
Screening of conventional PCR primers for *F. solani* where M is a 100 bp DNA ladder; Lanes 1 to 15 are primers FS-1/FS-2, FS-3/FS-4, F1-1/F1-2, F1-3/F1-4, F8-1/F8-2, F8-3/F8-4, F1/F2, F3/F4, F5/F6, F7/F8, F9/F10, F11/F12, F13/F14, F15/F16, and F17/F18.

**Figure 2 microorganisms-10-01037-f002:**
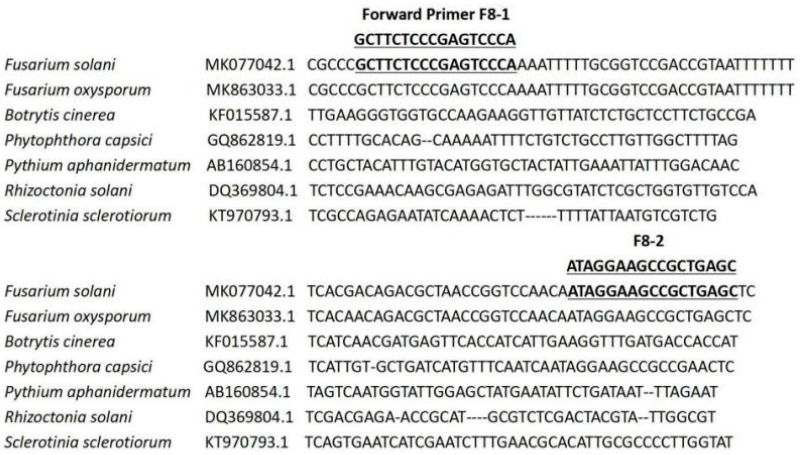
Multiple sequence alignment of the 189 bp EF gene region used as a qPCR target sequence for *F. solani.* (1) The *F. solani* strain with accession number MK077042.1; (2) the *F*. *oxysporum* strain with accession number MK863033.1; (3) the *B*. *cinerea* strain with accession number KF015587.1; (4) the *P*. *capsici* strain with accession number GQ862819.1; (5) the *P*. *aphanidermatum* strain with accession number AB160854.1; (6) the *R*. *solani* strain with accession number DQ369804.1; and (7) the *S*. *sclerotiorum* strain with accession number KT970793.1. The primer sequences are indicated (F8-1 is a forward complement and F8-2 is a reverse complement).

**Figure 3 microorganisms-10-01037-f003:**
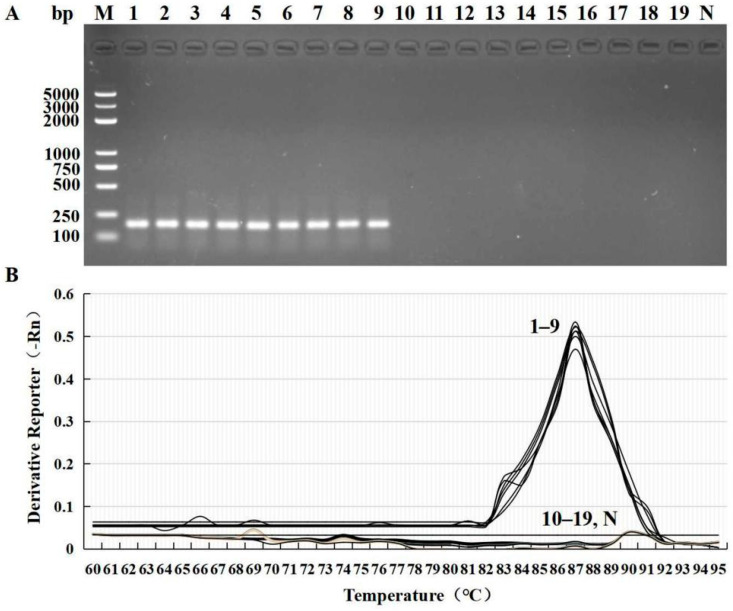
Specific detection of primers F8-1/F8-2 where M is a 100 bp DNA ladder (**A**), and the melting curve of the qPCR for gradient dilution of the genomic DNA of *F. solani* (**B**). Lane 1 is *F*. *solani*, Lane 2 is *F*. *oxysporum*, Lane 3 is *F*. *equiseti*, Lane 4 is *F*. *tricinctum*, Lane 5 is *F*. *avenaceum*, Lane 6 is *F*. *semitectum*, Lane 7 is *F*. *proliferatum*, Lane 8 is *F*. *acuminatum*, Lane 9 is *F*. *moniliforme*, Lane 10 is *P*. *aphanidermatum*, Lane 11 is *S*. *sclerotiorum*, Lane 12 is *P*. *infestans*, Lane 13 is *P*. *capsici*, Lane 14 is *B*. *cinerea*, Lane 15 is *S*. *solani*, Lanes 16–18 are *R*. *solani*, Lane 19 is *C*. *cassiicola*, and Lane N is the ddH_2_O blank.

**Figure 4 microorganisms-10-01037-f004:**
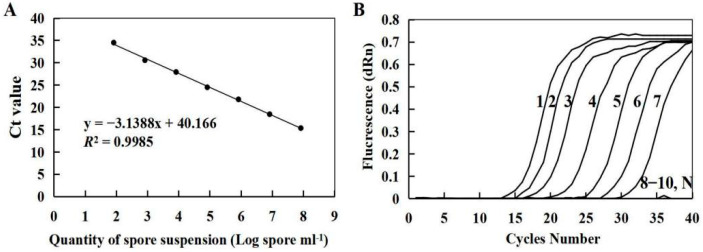
The standard curve (**A**) and amplification curve (**B**) of *F. solani*.

**Figure 5 microorganisms-10-01037-f005:**
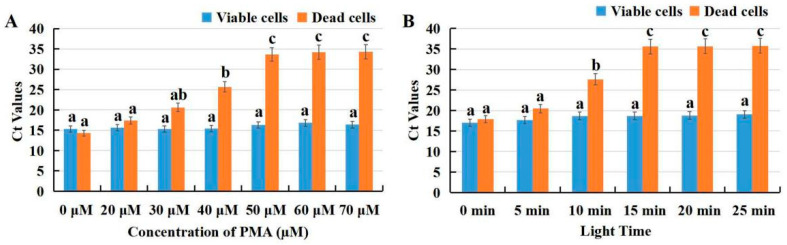
Effects of propidium monoazide (PMA) concentrations (**A**) and light exposure time (**B**) on the amplification of DNA from viable and dead *F. solani* cells at 10^7^ spores ml^−1^. Ct = cycle threshold. Columns labeled with different letters indicate statistically significant differences (*p* < 0.05).

**Table 1 microorganisms-10-01037-t001:** Fungal strains for the primers specificity test.

Species	Identification Method	Isolate Code ^a^	Host	Geographic Origin	Ct ^b^
*Fusarium solani*	microscopy	HG15051822	*Cucumis sativus*	Gansu	18.37
*Fusarium oxysporum*		HG17081402	*Cucumis sativus*	Henan	19.69
*Fusarium equiseti*		HG18042508	*Cucumis sativus*	Shouguang	24.42
*Fusarium tricinctum*		WJ18081532	*Lactuca sativa*	Gansu	21.67
*Fusarium avenaceum*		GL12052202	*Brassica oleracea*	Xizang	20.89
*Fusarium semitectum*		SG07121601	*Luffa cylindrical* Roem	Shouguang	25.45
*Fusarium proliferatum*		HG1004510	*Cucumis sativus*	Hebei	23.50
*Fusarium acuminatum*		LD1508081502	*Orobanche*	Xinjiang	25.88
*Fusarium moniliforme*		HG10052001	*Cucumis sativus*	Hebei	26.52
*Pythium aphanidermatum*		XG11012402	*Citrullus lanatus*	Hainan	>35.0
*Sclerotinia sclerotiorum*		KJ16022218	*Sonchus oleraceus*	Beijing	>35.0
*Phytophthora infestans*		FQ09092001	*Lycopersicon esculentum*	Gansu	>35.0
*Phytophthora capsici*		LJ12010802	*Capsicum annuum*	Hainan	>35.0
*Botrytis cinerea*		HG1603310804-1	*Cucumis sativus*	Ningxia	>35.0
*Stemphylium solani*		FQ14112206	*Lycopersicon esculentum*	Beijing	>35.0
*Rhizoctonia solani*		KXC1512180122	*Ipomoea aquatica* Forsk	Fujian	>35.0
*Rhizoctonia solani*		KXC151218030602	*Ipomoea aquatica* Forsk	Fujian	>35.0
*Rhizoctonia solani*		KXC151218031211	*Ipomoea aquatica* Forsk	Fujian	>35.0
*Corynespora cassiicola*		HG14061105	*Cucumis sativus*	Beijing	>35.0

^a^, type strain. ^b^, Ct value for 210 ng genomic DNA.

**Table 2 microorganisms-10-01037-t002:** Primers used for detecting *F. solani*.

Number	Primer Name	Sequence (5′-3′)	Fragment Size (bp)
1	FS-1	GCTTCTCCCGAGTCCCA	190
	FS-2	GCTCAGCGGCTTCCTATT	
2	FS-3	GCTTCTCCCGAGTCCCA	188
	FS-4	TCAGCGGCTTCCTATTGT	
3	F1-1	GCTTCTCCCGAGTCCCA	185
	F1-2	GCGGCTTCCTATTGTTG	
4	F1-3	GCTTCTCCCGAGTCCCA	191
	F1-4	AGCTCAGCGGCTTCCTA	
5	F8-1	GCTTCTCCCGAGTCCCA	189
	F8-2	GCTCAGCGGCTTCCTAT	
6	F8-3	GCTTCTCCCGAGTCCCA	165
	F8-4	CGGTTAGCGTCTGTTGTG	
7	F1	GCCTTGCTATTCCACATCG	240
	F2	GCTCAGCGGCTTCCTATT	
8	F3	GCGCCTTGCTATTCCAC	240
	F4	TCAGCGGCTTCCTATTGT	
9	F5	GAACCTCGCCTGGCATCT	218
	F6	TGGGACTCGGGAGAAGC	
10	F7	GCCTTGCTATTCCACATCG	238
	F8	TCAGCGGCTTCCTATTGT	
11	F9	GCCTTGCTATTCCACATCG	241
	F10	AGCTCAGCGGCTTCCTA	
12	F11	GCCTTGCTATTCCACATCG	152
	F12	GCAGGGATCAGGGCTTT	
13	F13	GTTGGACAAAGCCCTGAT	131
	F14	TTGAAGGAACCCTTACCG	
14	F15	GCTTCTCCCGAGTCCCA	165
	F16	CGGTTAGCGTCTGTTGTG	
15	F17	GTTGGACAAAGCCCTGAT	235
	F18	TGACGGTGACATAGTAGCG	

**Table 3 microorganisms-10-01037-t003:** Comparison of total and viable *F. solani* spore counts obtained by different methods ^a^.

Ratio ^b^	qPCR		PMA-qPCR		Colony Counting
	Ct Values	Total Spore Counts	Ct Values	Viable Spore Counts	Viable Spore Counts
100%	15.01 ± 0.53	8.01 ± 0.16 a	15.45 ± 0.45	7.87 ± 0.14 a	7.03 ± 6.98 a
75%	15.89 ± 0.11	7.73 ± 0.03 a	19.06 ± 0.12	6.72 ± 0.04 b	6.78 ± 1.41 b
50%	14.30 ± 0.37	8.24 ± 0.12 a	21.17 ± 0.36	6.05 ± 0.12 c	6.56 ± 3.68 b
25%	14.69 ± 0.93	8.12 ± 0.30 a	23.37 ± 0.50	5.35 ± 0.16 d	6.27 ± 1.25 bc
1%	15.32 ± 0.16	7.92 ± 0.05 a	25.85 ± 0.13	4.56 ± 0.04 e	5.43 ± 1.25 d
0.1%	15.08 ± 0.58	7.99 ± 0.19 a	29.09 ± 0.18	3.53 ± 0.06 f	4.82 ± 0.47 e

^a^ The total spore concentration (including viable and dead cells) was 1 × 10^7^ spores mL. Total and viable spore counts are shown at log spores per milliliter. ^b^ Ratio of viable spores. Different letters in the same column indicate statistically significant differences (*p* < 0.05).

**Table 4 microorganisms-10-01037-t004:** Determination of viable cells in soil samples artificially infected with *F. solani* by different methods ^a^.

Infected (Spore mL^−1^)	PMA-qPCR		Plate-Counting	Bioassay
	Ct Values	Viable Spore Counts	Viable Spore Counts	DI (%)
1 × 10^7^ viable spores	18.45 ± 0.44	8.77 × 10^6^	9.13 × 10^6^	92.67 ± 2.62
1 × 10^6^ viable spores	21.38 ± 0.76	1.13 × 10^6^	7.53 × 10^6^	76.00 ± 2.45
1 × 10^5^ viable spores	23.72 ± 0.22	1.76 × 10^5^	6.67 × 10^5^	47.33 ± 1.25
1 × 10^4^ viable spores	27.07 ± 0.60	1.62 × 10^4^	1.33 × 10^5^	10.67 ± 1.25
1 × 10^3^ viable spores	31.24 ± 0.45	7.36 × 10^2^	6.67 × 10^4^	0.00
1 × 10^2^ viable spores	34.06 ± 0.36	91.24	0.00	0.00
10 viable spores	>35	Not available	0.00	0.00
10^7^ dead spores	>35	Not available	0.00	0.00
0	>35	Not available	0.00	0.00

^a^ Viable spore counts are shown as spores per gram of soil. Not available means the *F. solani* spore counts were zero or below the detection limit.

**Table 5 microorganisms-10-01037-t005:** Detection of viable *F. solani* in naturally contaminated cucumber soil.

Sample	Location	Year	PMA-qPCR		Bioassay
			Ct Values	Viable Spore Counts	DI (%)
1	Beijing	2019	24.52 ± 0.24	9.65 × 10^4^	32.67 ± 2.49
2	Beijing	2019	23.87 ± 0.27	1.56 × 10^5^	35.33 ± 1.70
3	Shandong	2020	18.25 ± 0.30	9.60 × 10^6^	68.67 ± 3.40
4	Shandong	2020	20.15 ± 0.34	2.38 × 10^6^	58.67 ± 1.25
5	Shandong	2020	21.38 ± 0.27	9.66 × 10^5^	56.67 ± 2.49
6	Hebei	2020	26.75 ± 0.22	1.88 × 10^4^	15.33 ± 1.70
7	Hebei	2020	26.12 ± 0.21	2.99 × 10^4^	14.67 ± 3.30
8	Hebei	2020	24.78 ± 0.20	7.98 × 10^4^	16.00 ± 3.74
9	Beijing	2019	>35	Not available	0.00
10	Beijing	2019	>35	Not available	0.00
11	Beijing	2019	>35	Not available	0.00
12	Beijing	2019	>35	Not available	0.00
13	Beijing	2019	>35	Not available	0.00
14	Shandong	2020	>35	Not available	0.00
15	Shandong	2020	>35	Not available	0.00
16	Shandong	2020	>35	Not available	0.00
17	Hebei	2020	>35	Not available	0.00
18	Hebei	2020	>35	Not available	0.00

## Data Availability

The data presented in this study are available upon request from the corresponding author.

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
