# Peer review of "An Improved Method for Quantification of Viable Fusarium Cells in Infected Soil Products by Propidium Monoazide Coupled with Real-Time PCR"

_microorganisms, 2022, doi:10.3390/microorganisms10051037_

Round 1

Reviewer 1 Report

The manuscript is well written and presents a very interesting topic. The only shortcoming is the little discussion and conclusion of the work.

I recommend a revision of the English.

L13: Change “study aim was developed” with “study’s aim was to develop”.

L16,L60,L78,L85,L319,L326: Change TEF with EF. All the scientific paper use the acronym EF for the elongation factor gene.

L19,L20,L22,L106,L117,L119,L122,L124,L125,L127,L131,L140,L141,L143,L150,L154,L155,L201,L206,L207,L208,L217,L220,L221,L222,L224,L245,L247,L249,L254,L255,L258,L259,L260,L262,L263,L276,L287,L290,L293,L309,L310,L312,L313,L314,L330,L332,L333,L335,L336,L337: Change “mmol·L-1 “ with “mmol l-1 “. Check all the unit measurement inside the text and use the same format for all text.

L64: The Material and Methods could be write in past form.

L65: Change “Fungal Isolates, Cultivation, and DNA Extraction” with “Fungal isolates, cultivation, and DNA extraction”.

L68-L69: Change “potato dextrose agar” with “Potato Dextrose Agar”. Delete the sentence inside the brackets “containing per L of water: potato 200 g, dextrose 20 g and agar 20 g”.

L74: After “were stored at -40°C” add this information “for the future analysis.”.

L75: Change qPCR with primers. Within the table there are Ct values. If these values are obtained from the analyses performed in this article, they must be entered in the results section. When you have inserted the appendices at the bottom of the table, insert a single appendix for each row and delete the comma after each appendix.

L77: Change “Primer Design and Qpcr Conditions” with “Primer design and qPCR conditions”.

L78-L80: I did not understand what did you mean.

L83-L84: The fact that you only use F8-1/F8-2 among the chosen primers must be written in the results.

L87: I can't figure out whether the primers you listed in Table 2 were designed by you during these analyses or came from other studies.

L89, L93: The figure 1 and 2 must insert inside the results section.

L103,L152: Change analyzed with analysed.

L100-L103: I suggest to write the final concentration of mix, primers and template DNA and furthermore you forgot to describe the machine that you used to obtain the results and the thermoprofile. If the primers are yours, you must describing the thermoprofile.

L104: Change “Establishment of Standard Curve” with “Establishment of standard curve”.

L107-L109: Delete all the sentence. All the person know what it is the standard curve.

L115: Change “PMA Treatment” with “PMA treatment”.

L128: Change “The DNA extracted” with “The DNA was extracted”.

L129: Change “then 1 μL DNA” with “and in final 1 μl DNA”.

L135: Change “was repeated for 3 times” with “was prepared 3 replicates”.

L137: Change “Detection of Defined Ratio of Mixture Suspension” with “Detection of defined ratio of mixture suspension”.

L142: Change “three methods. i) Dilute the mixed suspension” with “three methods: i) dilute mixed suspension”.

L146: Change “Each treated samples was repeated for 3 times” with “For each treated samples was prepared 3 replicates”.

L147: Change “Detection of F. Solani From Artificially Contaminated Soil” with “Detection of F. solani from artificially contaminated soil”.

L153: Change “three methods. i) Mix 9 ml” with “three methods: i) mix 9 ml”.

L154: Change “ii) Pellets were” with “ii) pellets were”.

L155: What it mean PBS????

L156: Change “iii) For bioassays” with “iii) for bioassays”.

L161: Change “after 7 d” with “after 7 days”.

L165: Change “Detection of F. Solani from Naturally Infested Soil” with “Detection of F. solani from naturally infested soil”.

L166,L285,L288: Change lots with samples.

L169-L171: Change: “Sift the soil in different areas to remove excess plant residues. Extract 200 g of soil sample and 200 mL of sterile distilled water, mix them in equal proportion, and incubate them in a shaking table for 10 min (25°C in the dark)” with “The soil was sifted to remove excess plant residues. 200 g of soil sample and 200 ml of sterile distilled water were mixed in equal proportion, and incubate them in a shaking table for 10 min (25°C in the dark)”.

L173: Delete the sentence “The detection method is as described above”.

L174: Change “Data Analysis” with “Data analysis”.

L175: Add a “the” before Analysis.

L176: Change “Significance was determined” with “The significance value was”.

L177: Delete “the” before 95%.

L179: The results could be written in past tense.

L180: Change “Specificity of the Primers” with “Specificity of the primers”. Attach here the photo of the conventional PCR and the “match” of the primer with the sequence of F. solani.

L186: With figure 3a attached the table with the Ct.

L193: Inside the section 3.1, explain better why did you use the primer F-81/F8-2.

L200: Change “Sensitivity and Standard Curve of Qpcr” with “Sensitivity and standard curve of qPCR”.

L211: Figure 4 bad quality. Delete the sentence “In Figure A, a standard curve of the qPCR was generated using a 10-fold serial dilution of F. solani cell suspensions (8.2×107 212 spore·mL-1 to 82 spore·mL-1). In Figure B, No. 1 to No. 10 represent DNA concentrations ranging from 8.2×107 spore·mL-1 to 8.2×10-3 spore·mL-1.”.

L215: Change “PMA Concentration Optimization” with “PMA concentration optimization”.

L234: Figure 5 bad quality.

L236: Delete the sentence “The same below.”.

L244: Where is the unit of measurement of 15.45 ± 0.45 to 29.09 ± 0.18?

L246: Change shows with showed.

L252: Change “Detection of Soil Inoculated With F. Solani By Different Methods” with “Detection of soil inoculated with F. solani by different methods”.

L253: The sentence “For evaluate the applicability of PMA-qPCR method in detecting F. solani viable cells in soil.” did not have sense. Change it.

L254: Change “viable F. solani spore (107~10 spore·mL−1) and dead F. solani spores (107 spore·mL−1)” with “viable F. solani spore (107~10 spore·ml−1) and dead spores (107 spore·ml−1)”.

L267: The sentence “In the soil treatment infected with 104 to 107 viable spores of F. solani, the cucumber plants were infected” did not have sense. Change it.

L279: Delete it.

L280: Change “Detection of F. solani in Naturally Infested Soil” with “Detection of F. solani in Naturally Infested Soil”.

L281-L282: Delete the sentence “To detect the presence of F. solani viable spores in 18 naturally infected soil samples by PMA-qPCR method”.

L288: Change “(samples 1~2)” with “(samples 1 and 2)”.

L290: Delete “by PMA-qPCR”.

L300: Inside the table you need to write all the 18 samples.

L301: The discussion could be write in present form.

L321: Add the reference of the figure after “fungal species”.

L330,L332: Change cfu with CFU.

L332: Change “For fungi (Alternaria spp.) in vegetable industry” with “For Alternaria spp. in vegetable industry”.

L342: Change showed with shows.

L365: Check all the reference, it is not write correct. Author 1, A. B.; Author 2, C. D. Title of the article. Abbreviation Journal Year, Volume, Page range.

Author Response

Response to Reviewer 1 Comments

Point 1: I recommend a revision of the English.

Response 1: The language of the paper is completed by AJE company.

Point 2: L13: Change “study aim was developed” with “study’s aim was to develop”.

Response 2: Done.

Point 3: L16,L60,L78,L85,L319,L326: Change TEF with EF. All the scientific paper use the acronym EF for the elongation factor gene.

Response 3: Done.

Point 4: L19,L20,L22,L106,L117,L119,L122,L124,L125,L127,L131,L140,L141,L143,L150,L154,L155,L201,L206,L207,L208,L217,L220,L221,L222,L224,L245,L247,L249,L254,L255,L258,L259,L260,L262,L263,L276,L287,L290,L293,L309,L310,L312,L313,L314,L330,L332,L333,L335,L336,L337: Change “mmol·L-1 “ with “mmol l-1 “. Check all the unit measurement inside the text and use the same format for all text.

Response 4: Done.

Point 5: L64: The Material and Methods could be write in past form.

Response 5: Done.

Point 6: L65: Change “Fungal Isolates, Cultivation, and DNA Extraction” with “Fungal isolates, cultivation, and DNA extraction”.

Response 6: Done.

Point :7: L68-L69: Change “potato dextrose agar” with “Potato Dextrose Agar”. Delete the sentence inside the brackets “containing per L of water: potato 200 g, dextrose 20 g and agar 20 g”.

Response 7: Done.

Point 8: L74: After “were stored at -40°C” add this information “for the future analysis.”.

Response 8: Done.

Point 9: L75: Change qPCR with primers. Within the table there are Ct values. If these values are obtained from the analyses performed in this article, they must be entered in the results section. When you have inserted the appendices at the bottom of the table, insert a single appendix for each row and delete the comma after each appendix.

Response 9: Done.

Point 10: L77: Change “Primer Design and Qpcr Conditions” with “Primer design and qPCR conditions”.

Response 10: Done.

Point 11: L78-L80: I did not understand what did you mean.

Response 11: Only one primer is designed, which does not guarantee that the subsequent primer specificity, sensitivity and quality detection are qualified and applicable. The probability of success is small if only one primer is designed. Therefore, multiple primers are designed before the test, and more excellent specific primers are selected for the next test.

Point 12: L83-L84: The fact that you only use F8-1/F8-2 among the chosen primers must be written in the results.

Response 12: Done.

Point 13: L87: I can't figure out whether the primers you listed in Table 2 were designed by you during these analyses or came from other studies.

Response 13: The primers in Table 2 were designed by ourselves through primer design software.

Point 14: L89, L93: The figure 1 and 2 must insert inside the results section.

Response 14: Done.

Point 15: L103,L152: Change analyzed with analysed.

Response 15: Done.

Point 16: L100-L103: I suggest to write the final concentration of mix, primers and template DNA and furthermore you forgot to describe the machine that you used to obtain the results and the thermoprofile. If the primers are yours, you must describing the thermoprofile.

Response 16: Done.

Point 17: L104: Change “Establishment of Standard Curve” with “Establishment of standard curve”.

Response 17: Done.

Point 18: L107-L109: Delete all the sentence. All the person know what it is the standard curve.

Response 18: Done.

Point 19: L115: Change “PMA Treatment” with “PMA treatment”.

Response 19: Done.

Point 20: L128: Change “The DNA extracted” with “The DNA was extracted”.

Response 20: Done.

Point 21: L129: Change “then 1 μL DNA” with “and in final 1 μl DNA”.

Response 21: Done.

Point 22: L135: Change “was repeated for 3 times” with “was prepared 3 replicates”.

Response 22: Done.

Point 23: L137: Change “Detection of Defined Ratio of Mixture Suspension” with “Detection of defined ratio of mixture suspension”.

Response 23: Done.

Point 24: L142: Change “three methods. i) Dilute the mixed suspension” with “three methods: i) dilute mixed suspension”.

Response 24: Done.

Point 25: L146: Change “Each treated samples was repeated for 3 times” with “For each treated samples was prepared 3 replicates”.

Response 25: Done.

Point 26: L147: Change “Detection of F. Solani From Artificially Contaminated Soil” with “Detection of F. solani from artificially contaminated soil”.

Response 26: Done.

Point 27: L153: Change “three methods. i) Mix 9 ml” with “three methods: i) mix 9 ml”.

Response 27: Done.

Point 28: L154: Change “ii) Pellets were” with “ii) pellets were”.

Response 28: Done.

Point 29: L155: What it mean PBS????

Response 29: Done. phosphate buffer saline (PBS).

Point 30: L156: Change “iii) For bioassays” with “iii) for bioassays”.

Response 30: Done.

Point 31: L161: Change “after 7 d” with “after 7 days”.

Response 31: Done.

Point 32: L165: Change “Detection of F. Solani from Naturally Infested Soil” with “Detection of F. solani from naturally infested soil”.

Response 32: Done.

Point 33: L166,L285,L288: Change lots with samples.

Response 33: Done.

Point 34: L169-L171: Change: “Sift the soil in different areas to remove excess plant residues. Extract 200 g of soil sample and 200 mL of sterile distilled water, mix them in equal proportion, and incubate them in a shaking table for 10 min (25°C in the dark)” with “The soil was sifted to remove excess plant residues. 200 g of soil sample and 200 ml of sterile distilled water were mixed in equal proportion, and incubate them in a shaking table for 10 min (25°C in the dark)”.

Response 34: Done.

Point 35: L173: Delete the sentence “The detection method is as described above”.

Response 35: Done.

Point 36: L174: Change “Data Analysis” with “Data analysis”.

Response 36: Done.

Point 37: L175: Add a “the” before Analysis.

Response 37: Done.

Point 38: L176: Change “Significance was determined” with “The significance value was”.

Response 38: Done.

Point 39: L177: Delete “the” before 95%.

Response 39: Done.

Point 40: L179: The results could be written in past tense.

Response 40: Done.

Point 41: L180: Change “Specificity of the Primers” with “Specificity of the primers”. Attach here the photo of the conventional PCR and the “match” of the primer with the sequence of F. solani.

Response 41: Done. The result part completes the adjustment.

Point 42: L186: With figure 3a attached the table with the Ct.

Response 42: Done.

Point 43: L193: Inside the section 3.1, explain better why did you use the primer F-81/F8-2.

Response 43: Done.

Point 44: L200: Change “Sensitivity and Standard Curve of Qpcr” with “Sensitivity and standard curve of qPCR”.

Response 44: Done.

Point 45: L211: Figure 4 bad quality. Delete the sentence “In Figure A, a standard curve of the qPCR was generated using a 10-fold serial dilution of F. solani cell suspensions (8.2×107 212 spore·mL-1 to 82 spore·mL-1). In Figure B, No. 1 to No. 10 represent DNA concentrations ranging from 8.2×107 spore·mL-1 to 8.2×10-3 spore·mL-1.”.

Response 45: Done.

Point 46: L215: Change “PMA Concentration Optimization” with “PMA concentration optimization”.

Response 46: Done.

Point 47: L234: Figure 5 bad quality.

Response 47: Done.

Point 48: L236: Delete the sentence “The same below.”.

Response 48: Done.

Point 49: L244: Where is the unit of measurement of 15.45 ± 0.45 to 29.09 ± 0.18?

Response 49: No units are needed here. According to the literature description of Chai et al.(2020)

Point 50: L246: Change shows with showed.

Response 50: Done.

Point 51: L252: Change “Detection of Soil Inoculated With F. Solani By Different Methods” with “Detection of soil inoculated with F. solani by different methods”.

Response 51: Done.

Point 52: L253: The sentence “For evaluate the applicability of PMA-qPCR method in detecting F. solani viable cells in soil.” did not have sense. Change it.

Response 52: Done. I've finished delete.

Point 53: L254: Change “viable F. solani spore (107~10 spore·mL−1) and dead F. solani spores (107 spore·mL−1)” with “viable F. solani spore (107~10 spore·ml−1) and dead spores (107 spore·ml−1)”.

Response 53: Done.

Point 54: L267: The sentence “In the soil treatment infected with 104 to 107 viable spores of F. solani, the cucumber plants were infected” did not have sense. Change it.

Response 54: Done. I've finished delete.

Point 55: L279: Delete it.

Response 55: Done.

Point 56: L280: Change “Detection of F. solani in Naturally Infested Soil” with “Detection of F. solani in Naturally Infested Soil”.

Response 56: Done.

Point 57: L281-L282: Delete the sentence “To detect the presence of F. solani viable spores in 18 naturally infected soil samples by PMA-qPCR method”.

Response 57: Done.

Point 58: L288: Change “(samples 1~2)” with “(samples 1 and 2)”.

Response 585: Done.

Point 59: L290: Delete “by PMA-qPCR”.

Response 59: Done.

Point 60: L300: Inside the table you need to write all the 18 samples.

Response 60: Done.

Point 2: L301: The discussion could be write in present form.

Response 61: Done.

Point 2: L321: Add the reference of the figure after “fungal species”.

Response 61: Done.

Point 62: L330,L332: Change cfu with CFU.

Response 62: Done.

Point 63: L332: Change “For fungi (Alternaria spp.) in vegetable industry” with “For Alternaria spp. in vegetable industry”.

Response 63: Done.

Point 64: L342: Change showed with shows.

Response 64: Done.

Point 65: L365: Check all the reference, it is not write correct. Author 1, A. B.; Author 2, C. D. Title of the article. Abbreviation Journal YearVolume, Page range.

Response 65: Done.

Reviewer 2 Report

Lida Chen and colleagues submitted an article on the PMA-qPCR detection on viable Fusarium solani cells from infected soil samples. The topic is of interest to the readership of Microorganisms. However, some issue need to be clarified before the publication of the study.

The title is misleading since only data for viable F. solani cells from soils are provided in the manuscript.

Introduction: From the text, it is not quite clear why the authors focus on F. solani. Maybe it is better to start in line 31 with the information that F. solani is a severe danger for cucumber cultivation/add some information on losses. Why didn´t the authors include Fusarium oxysporum f. sp. cucumerinum in their PMA-qPCR assay? There is not only F. solani in the soils.

The authors provide a table with the origin of the fungal strains they used in their study and an isolate code. Is there a database behind these codes? Are there any sequence data provided for the strains to ensure their identity or how was this tested? Please add some information on this.

Line 79: Please add also information that the partial sequence of the TEF gene (MK077042) can be found at NCBI database.

Line 169-172. Please revise this sentence, it sounds like a copy+ paste phrase from a manual.

Maybe I missed it, but how were PMA-treatment and qPCR done on the soil samples? Did the authors used the same parameters and the same DNA extraction kit?

The resolution quality of the figures is too low for publication; I could hardly read the legends in Figure 4 and 5. Please improve the quality. Please delete the last sentence in Figure caption 5.  

Please also check the references 2, 14, 18, 30, 39 for spelling mistakes.

Author Response

Response to Reviewer 2 Comments

Lida Chen and colleagues submitted an article on the PMA-qPCR detection on viable Fusarium solani cells from infected soil samples. The topic is of interest to the readership of Microorganisms. However, some issue need to be clarified before the publication of the study.

Point 1: The title is misleading since only data for viable F. solani cells from soils are provided in the manuscript.

Response 1: Thank you for your questions. I think it is more appropriate, because PMA can inhibit the expansion of dead cells, and living cells are of certain significance to our disease research. In addition, according to the literature, the topics described by Chai et al (2020), Meng et al (2016),Abdolali Golpayegani et al (2019), Carolinne Odebrecht Dias et al (2020) and others are similar and consistent with the theme of the paper.

Point 2: Introduction: From the text, it is not quite clear why the authors focus on F. solani. Maybe it is better to start in line 31 with the information that F. solani is a severe danger for cucumber cultivation/add some information on losses. Why didn´t the authors include Fusarium oxysporum f. sp. cucumerinum in their PMA-qPCR assay? There is not only F. solani in the soils.

Response 2: Done. 1. I have completed the severity of the disease in cucumber cultivation. At present, it is reported in the literature that the primers for the detection of F. solani and Fusarium oxysporum are difficult to distinguish, and the sequence similarity is strong. Therefore, designing primers for F. solani can detect the content of all Fusarium, which saves the working time of designing primers for all Fusarium.

Point 3: The authors provide a table with the origin of the fungal strains they used in their study and an isolate code. Is there a database behind these codes? Are there any sequence data provided for the strains to ensure their identity or how was this tested? Please add some information on this.

Response 3: These codes are not databases, but isolation numbers of strains. All fungal strains were confirmed by microscopic examination.

Point 4: Line 79: Please add also information that the partial sequence of the TEF gene (MK077042) can be found at NCBI database.

Response 4: Done.

Point 5: Line 169-172. Please revise this sentence, it sounds like a copy+ paste phrase from a manual.

Response 5: Done.

Point 6: Maybe I missed it, but how were PMA-treatment and qPCR done on the soil samples? Did the authors used the same parameters and the same DNA extraction kit?

Response 6: PMA treatment of soil in method 2.4. qPCR method is to detect the extracted soil DNA (without soil pretreatment). We used the same parameters and the same DNA extraction Kit (except PMA test).

Point 7: The resolution quality of the figures is too low for publication; I could hardly read the legends in Figure 4 and 5. Please improve the quality. Please delete the last sentence in Figure caption 5.  

Response 7: Done. We have remade the picture to improve the quality of the picture

Point 8: Please also check the references 2, 14, 18, 30, 39 for spelling mistakes.

Response 8: Done. No spelling mistakes.